# Inhibition of Wnt/β-Catenin Signaling Sensitizes Esophageal Cancer Cells to Chemoradiotherapy

**DOI:** 10.3390/ijms221910301

**Published:** 2021-09-24

**Authors:** Melanie Spitzner, Georg Emons, Karl Burkhard Schütz, Hendrik A. Wolff, Stefan Rieken, B. Michael Ghadimi, Günter Schneider, Marian Grade

**Affiliations:** 1Department of General, Visceral and Pediatric Surgery, University Medical Center Goettingen, 37075 Goettingen, Germany; melanie.spitzner@med.uni-goettingen.de (M.S.); georg.emons@med.uni-goettingen.de (G.E.); karlburkhard.schuetz@sanktgeorg.de (K.B.S.); mghadim@uni-goettingen.de (B.M.G.); guenter.schneider@med.uni-goettingen.de (G.S.); 2Department of Urology and Andrology, Sankt Georg Medical Centre and Hospital, 04129 Leipzig, Germany; 3Department of Radiotherapy and Radiooncology, University Medical Center Goettingen, 37075 Goettingen, Germany; h.wolff@strahlentherapie-muenchen.eu (H.A.W.); stefan.rieken@med.uni-goettingen.de (S.R.); 4Department of Radiology, Nuclear Medicine and Radiotherapy, Radiology Munich, 80331 Munich, Germany

**Keywords:** esophageal cancer, chemoradiotherapy, treatment resistance, chemoradiotherapy-sensitization, Wnt/β-catenin pathway, tankyrase inhibition

## Abstract

The standard treatment of locally advanced esophageal cancer comprises multimodal treatment concepts including preoperative chemoradiotherapy (CRT) followed by radical surgical resection. However, despite intensified treatment approaches, 5-year survival rates are still low. Therefore, new strategies are required to overcome treatment resistance, and to improve patients’ outcome. In this study, we investigated the impact of Wnt/β-catenin signaling on CRT resistance in esophageal cancer cells. Experiments were conducted in adenocarcinoma and squamous cell carcinoma cell lines with varying expression levels of Wnt proteins and Wnt/β-catenin signaling activities. To investigate the effect of Wnt/β-catenin signaling on CRT responsiveness, we genetically or pharmacologically inhibited Wnt/β-catenin signaling. Our experiments revealed that inhibition of Wnt/β-catenin signaling sensitizes cell lines with robust pathway activity to CRT. In conclusion, Wnt/β-catenin activity may guide precision therapies in esophageal carcinoma patients.

## 1. Introduction

Esophageal cancer is a leading cause of cancer-related morbidity and mortality, with 19,260 estimated new cases and 15,530 estimated deaths in the US in 2021, and 5-year overall survival rates of around 20% [1]. In locally advanced stages of the disease, the standard treatment typically involves a multidisciplinary approach. Depending on the underlying histology (squamous-cell carcinoma (SCC) or adenocarcinoma (AC)), and factors such as localization of the tumor and comorbidities of the patient, either preoperative chemoradiotherapy (CRT) or perioperative chemotherapy (CT) are applied, both followed by radical surgical resection, or definitive CRT without surgery [2,3]. However, the response to pre- or perioperative treatment modalities is heterogeneous, and ranges from complete histopathological regression to complete resistance. This has resulted in significant efforts and clinical trials investigating different multimodal treatment strategies [4,5,6]. A prime example is the CROSS trial, which demonstrated that the median overall survival (OS) was significantly higher in the CRT-plus-surgery group compared to the surgery-only group (48.6 months versus 24.0 months, respectively) [7,8]. Very recently, the follow-up data from the CROSS trial were published, with a 10-year OS of 38% for patients in the CRT-plus-surgery group compared with 25% for patients in the surgery-only group, respectively [9]. Nevertheless, many patients do not benefit from the advances in current treatment strategies, but are exposed to the potential side effects of both CT and irradiation. Accordingly, alternative concepts for patient stratification and novel treatment strategies are urgently needed [10,11]. This is particularly important as patients with preoperative treatment concepts, or relevant comorbidities who are unable to undergo surgical resection, highly rely on the efficacy of CRT, as this represents the only therapeutic choice.

Unfortunately, the molecular basis of the heterogeneous response largely remains unknown. Accordingly, there is a strong need to elucidate the molecular characteristics underlying treatment resistance in esophageal cancer, and to identify alternative strategies to increase the fraction of patients who respond to multimodal treatment. Depending on the histological subtype, esophageal cancer frequently harbors copy number alterations or mutations in oncogenes or tumor suppressor genes like TP53, SMAD4, ERBB2, NOTCH1, GATA4, KLF5, PIK3CA, and CTNNB1 [12,13,14]. Furthermore, several cellular signaling pathways are commonly deregulated, including PTK/PI3K/AKT, JAK/STAT, MAPK, and NF-κB [14,15,16]. Modulation of those genes or pathways in combination with CRT regimens could help to develop novel strategies. Such a note is underscored by clinical data, e.g., the randomized controlled “Trastuzumab for Gastric Cancer (ToGA)” trial for patients with unresectable or metastasized gastric and gastro-esophageal junction cancer [17].

Increasing evidence indicates that Wnt/β-catenin signaling plays an important role for the development and progression of esophageal cancer [18,19,20,21,22]. In this context, our group previously demonstrated that Wnt/β-catenin signaling mediates CRT resistance of colorectal cancer (CRC) cells accompanied by a compromised DNA double strand break repair [23,24]. In the present study, we evaluated whether Wnt/β-catenin signaling has a functional role in resistance of esophageal cancer cells to CRT. We show that active basal Wnt/β-catenin signaling is associated with CRT resistance. Furthermore, interfering with Wnt/β-catenin signaling re-sensitizes esophageal cancer cells to CRT, suggesting a potential opportunity for future treatment strategies.

## 2. Results

### 2.1. Esophageal Cancer Cell Lines Show Different Wnt/β-Catenin Pathway Activities

We and others have demonstrated that aberrant Wnt/β-catenin signaling plays an important role in mediating resistance to different treatment modalities, including CRT [23,24,25,26]. To assess the influence of Wnt/β-catenin signaling in esophageal cancer cells, we first determined the expression levels of relevant Wnt-related proteins such as Axin2, active β-catenin (ABC), total β-catenin (TBC), and the Wnt-transcription factor TCF7L2 by Western blot analysis (Figure 1A). Since there are two main histological subtypes of esophageal cancer, five AC and four SCC cell lines were included in this study. Figure 1A shows that the expression levels of Wnt-related proteins differ among the cell lines, with relatively high expression of active and total β-catenin in the AC cell lines OAC-P4C, SK-GT4, and FLO-1, and the SCC cell lines Kyse-180 and Kyse-150. The Wnt-transcription factor TCF7L2 is expressed in all cell lines, but with varying intensities. The variable expression levels of Wnt-related proteins prompted us to investigate a different transcriptional output of the Wnt/β-catenin pathway. Therefore, we applied TOP*Flash*/FOP*Flash* reporter assays to assess the Wnt/β-catenin activity represented by the transcriptional TCF/LEF reporter activity, as a standard in the field [27]. Due to low transfection efficacy, we were not able to analyze Kyse-70 and Kyse-270 cells. Basal Wnt/β-catenin activity was detected in two AC cell lines, OE-19 and OE-33 (Figure 1B). OAC-P4C, FLO-1, Kyse-150, and Kyse-180 showed no basal activity (Figure 1B). To test if exogenous induction of Wnt/β-catenin activity triggers a transcription reporter signal, we assessed reporter activity after transfection of an expression plasmid encoding for a mutated, constitutively active version of β-catenin (S33Y), which cannot be degraded by GSK3β phosphorylation [28]. As displayed in Figure 1C, the high basal TCF/LEF reporter activity of OE-19 cells was further increased (800-fold). Except for OAC-P4C, the other cell lines showed inducible reporter activity.

To assess responsiveness to RT, either alone or in combination with 5-FU, we next performed colony formation assays (CFA), as standard in the field. This assay measures the ability of a single cell to form a colony, while one colony is defined as at least 50 living cells [29]. Towards this goal, we first determined 5-FU sensitivities using doses ranging from 0.1 to 50 µM (Figure 1D,E), since 5-FU is a documented radiosensitizer [30,31]. In accordance with our established protocol for CRC cell lines [32], we selected a concentration of 3 µM 5-FU. To determinate the respective sensitivities to RT and CRT, all cell lines were subjected to CFA analyses (Figure 1F,G). The linear-quadratic model (LQ model) was used to quantify the effects of radiation and/or radiosensitizing agents [33]. As a result, we observed that the survival rates following RT or CRT varying between the cell lines. OAC-P4C, FLO-1, and Kyse-150 were highly resistant, whereas SK-GT4, Kyse-70, and Kyse-180 were relatively sensitive (Figure 1F,G). Comparing RT and CRT effects in the same cell line, there was a benefit for OE-19, OE-33, SK-GT-4, Kyse-70, and Kyse-180 through the addition of 5-FU (Figure 1H,I). Next, we aimed to investigate the effect of Wnt/β-catenin signaling inhibition in a CRT setting.

### 2.2. RNAi-Mediated Inhibition of Wnt/β-Catenin Signaling Sensitizes to CRT

Beta-catenin is the key intracellular signal transducer of canonical Wnt signaling, which, in the absence of Wnt-ligands, is targeted for degradation by a multi-protein destruction complex consisting mainly of adenomatous polyposis coli (APC), Axin, glycogen synthase kinase 3 beta (GSK3β), and casein kinase 1 alpha (CK1α) [34,35]. Pathway activation occurs through ligand-receptor binding followed by the release of β-catenin from the disintegrated destruction complex. Stabilized β-catenin accumulates in the cytosol, translocates to the nucleus, and subsequently binds to the transcription factors T cell factor (TCF) and lymphoid enhancer–binding factor 1 (LEF1), which in turn up-regulates β-catenin/TCF target genes [35,36,37]. Due to its function as key signal regulator of the canonical Wnt pathway, we silenced β-catenin using RNAi. Depletion of β-catenin resulted in a suppression of both active and total β-catenin protein levels (Figure 2A–F, upper left). Of note, active β-catenin was inconsistently detected in OE-33 cells, which may be due to the lower expression levels. RNAi-mediated silencing of β-catenin had no effect on either cellular viability (Figure 2A–F, upper right), or plating efficiency. As shown in Figure 2A,B, treatment with CRT after RNAi resulted in a significant reduction of clonogenic survival in Wnt/β-catenin active OE-19 cells (RER_siRNA#1_ = 1.37, RER_siRNA#2_ = 1.24 at 4 Gy), and in OE-33 cells (RER_siRNA#1_ = 1.32, RER_siRNA#2_ = 1.37 at 4 Gy). In contrast, there was no effect in OAC-P4C, FLO-1, Kyse-150, and Kyse-180 cells (Figure 2C–F). These results suggest that the effect of CRT sensitization after pathway inhibition is restricted to AC cells with basal transcriptional Wnt/β-catenin activity.

### 2.3. Fractionated Irradiation in Wnt/β-Catenin Signaling-Independent Cell Lines

In the clinical setting, irradiation is delivered in fractionated doses [38]. Therefore, we mimicked this strategy and tested Wnt/β-catenin inhibition in the context of a fractionated CRT setting. Because 1.8–2 Gy represents the typical individual dose of conventional fractionation delivery RT [39], we used a single dose of 2 Gy every 12 h until a total dose of 10 Gy (Appendix A). Prior irradiation, cells were transfected with siRNA. We only implemented this protocol for those four cell lines for which we did not observe a re-sensitization effect, i.e., OAC-P4C, FLO-1, Kyse-150, and Kyse-180 (Figure 2C–F). Successful silencing following RNAi against β-catenin was assessed after each fraction time point, and analyzed by Western blotting (Figure 3A–D, left). As expected, the CRT sensitivity of OAC-P4C, with no basal and no inducible signaling reporter activity (Figure 1B,C), remained unchanged (Figure 3A). Likewise, FLO-1 and Kyse-180, both of which harbor inducible Wnt/β-catenin activity (Figure 1C), revealed no changes in the CRT survival (Figure 3B,C), respectively. As displayed in Figure 3D, the highly resistant SCC cell line Kyse-150 revealed impaired clonogenic survival following depletion of β-catenin in a fractionated treatment regimen (RER_siRNA#1_ = 1.6, RER_siRNA#2_ = 1.41 at 6 Gy). However, the surviving fraction of Kyse-150 cells remains high with a fractionated irradiation protocol after β-catenin inhibition, which limits the value of targeting the Wnt/β-catenin pathway in this model. At this point, it is important to note that Kyse-150 originates from a patient who was treated with RT [40]. Therefore, this cell line is highly refractory to CRT.

### 2.4. Using Small-Molecule Inhibitors as a Therapeutic Strategy to Sensitize Esophageal Cancer Cells

To evaluate whether targeting Wnt/β-catenin signaling at a pharmacological level represents a potential clinical strategy for Wnt/β-catenin active tumors, we tested XAV-939 [41]. This compound is a small-molecule inhibitor of tankyrase 1 and tankyrase 2, which stabilize Axin as part of the β-catenin destruction complex [35,41]. First, we determined effective concentrations and inhibition time points to assess further functional irradiation assays in all six cell lines. For OE-19, OE-33, OAC-P4C, FLO-1, and Kyse-180, two doses of XAV-939 were successfully established. Western blot analyses demonstrated induction of Axin2 expression and, at the same time, reduction of active and total β-catenin (Figure 4A–E, upper left). In OE-19, OAC-P4C, FLO-1, and Kyse-180 cells, cellular viability was significantly reduced following XAV-939 treatment (Figure 4A–E, upper right). However, this reduction of cellular viability had no effect on colony formation and did not limit irradiation experiments. The most striking effect of XAV-939 treatment on survival after CRT was observed in OE-19 cells, with an RER_2.5 µM XAV-939_ = 1.61 and an RER_5 µM XAV-939_ = 1.56 at 4 Gy (Figure 4A). This effect was even higher compared to the effect observed after CRT following RNAi-mediated depletion of β-catenin (Figure 2A). In Wnt/β-catenin active OE-33 cells, treatment with XAV-939 resulted in decreased CRT survival rates with an RER_5 µM XAV-939_ = 1.39 and an RER_10 µM XAV-939_ = 1.23 at 4 Gy (Figure 4B). In none of the other cell lines (Figure 4C–E), CFA survival was affected after treatment with XAV-939.

To further confirm these results, we used JW55, another tankyrase inhibitor. In contrast to XAV-939, this small-molecule inhibitor can be administered orally [42]. Again, we first established reasonable doses and time points for JW55. Upon treatment with two doses of JW55, OE-19, OE-33, FLO-1, Kyse-150, and Kyse-180 showed increased Axin2 levels and reduced active and total β-catenin protein expression (Figure 5A–E, upper left). In general, cellular viability of JW55 treated cells was not affected when compared to the DMSO control (Figure 5B–E, upper right). Only OE-19 cells revealed an impaired cellular viability (Figure 5A upper right), which did not interfere with the ability to form colonies. In clonogenic survival assays, treatment of OE-19 and OE-33 cells with JW55 rendered both cell lines more sensitive to CRT, as revealed by their decreased CFA survival rates with an RER_5 µM JW55_ = 1.29 and an RER_10 µM JW55_ = 1.54 at 4 Gy for OE-19 (Figure 5A) and an RER_10 µM JW55_ = 1.39 at 4 Gy for OE-33 (Figure 5B). Note that none of the JW55 concentrations affected CFA survival of Wnt/β-catenin inactive cell lines FLO-1, Kyse-150, and Kyse-180 (Figure 5C–E). In summary, only cell lines with relevant basal transcriptional Wnt/β-catenin activity could be (re-) sensitized to CRT upon tankyrase/β-catenin inhibition, either by XAV-939 or JW55.

## 3. Discussion

Treatment resistance represents a fundamental problem in clinical oncology. In this context, esophageal cancer represents a prime example, because success rates remain low with dismal outcomes and 5-year survival rates ranging from 20% to 38% [1,7,9], indicating that for only a subset of patients, multimodal treatment concepts are effective [10,11].

In our set of esophageal cancer cell lines, we observed for both histological subtypes (SCC and AC) different response rates to RT and 5-FU-based CRT, measured by clonogenic survival. Comparing CRT responses of these cell lines (Figure 1F,G) to our observations in a panel of CRC cell lines [32], esophageal cancer cell lines appear to be more resistant to CRT. In our study, we showed that for a subset of cell lines with high basal and inducible Wnt/β-catenin signaling activity, sensitization to CRT upon inhibition of the pathway occurs. Inhibition of Wnt/β-catenin signaling was achieved by different methods (RNAi and pharmacological blockade), which resulted in re-sensitization of the resistant AC lines OE-19 and OE-33 to CRT. Since, for technical reasons, we were only able to assess basal activity in two SCC lines, both of which revealed no Wnt/β-catenin reporter activity, we currently cannot conclude the potential role of Wnt/β-catenin pathway inhibition in this subtype. However, other groups have investigated radiation resistance in SCC esophageal cancer cells, and provided evidence for a potential involvement of Wnt/β-catenin signaling in mediating RT resistance in this histological subtype [43,44,45,46]. In one study, the authors used an indirect approach to inhibit Wnt/β-catenin signaling via a microRNA, which influences the expression of Wnt/β-catenin-related genes. miRNA-381 was found to be downregulated in resistant SCC tissues and in cell lines exhibiting a re-sensitizing effect after expression, whereas inhibition thereof, promoted radiation resistance. However, this study was not primarily focused on Wnt/β-catenin signaling, it was rather a screen of microRNAs to compare tissues from primary esophageal SCC and recurrent esophageal SCC following RT [46]. In another study of esophageal SCC, two isogenic radioresistant cell lines were generated and showed changes in the expression levels of nuclear β-catenin and c-myc, which resulted in an enhanced RT resistance compared to the corresponding parental cells [47]. Mechanistically, it was demonstrated that Wnt/β-catenin signaling promotes DNA damage repair by transactivation of the high-mobility group box 1 protein (HMGB1) [47], an observation that is consistent with our investigation of radiation resistant CRC cells [24]. One limitation of this study is that the authors determined Wnt/β-catenin activity only by Western blot analysis and IF staining of β-catenin and c-myc, instead of measuring Wnt/β-catenin activity by TOP*Flash*/FOP*Flash* reporter assays, as standard in the field [27]. In this context, we observed that there was no correlation between β-catenin protein expression and basal Wnt/β-catenin activity, as shown for OE-19 and OE-33 cells. The molecular reasons are still unclear and demand further experimentation. Together, our data and these studies point to a potential role for inhibiting Wnt/β-catenin signaling as a therapeutic concept to increase responsiveness of esophageal cancer to CRT.

The potential relevance of Wnt/β-catenin signaling in mediating RT/CRT resistance has been demonstrated in other tumor entities, including CRC [23,24,48,49], prostate cancer [50], lung cancer [51], head and neck cancer [52], breast cancer and mammary gland cells [53], nasopharyngeal cancer [54], glioblastoma [55], and pancreatic cancer [56]. Although these reports underpin the relevance of Wnt/β-catenin signaling for radioresistance, the underlying mechanisms are still not fully understood. Wnt/β-catenin signaling triggers numerous cellular and molecular mechanisms presumably involved in drug efflux, DNA damage repair, inhibition of apoptosis, regulation of the cell cycle, cellular survival, reactive oxygen species (ROS), induction of epithelial to mesenchymal transition (EMT), and modification of the tumor microenvironment (TME) [25,26,57], which all can be connected to treatment resistance.

Until today, extensive efforts have been made in the development of small-molecule inhibitors that target the Wnt/β-catenin pathway, but none of them have yet reached clinical application as an FDA approved drug [34,35,49,58]. However, several inhibitors entered clinical testing, which include OMP-18R5 (vantictumab), a monoclonal antibody against Frizzled receptors, OMP-54F28, which binds to all Wnt-ligands, LGK974 and ETC-1922159 as examples for porcupine inhibitors preventing the production of bioactive Wnt-ligands [49]. Novel pharmacological concepts, which allow direct degradation of β-catenin, are under development [59]. Such data underscore the translational potential of our study.

From our data, only a subset of esophageal cancer cell lines showed basal Wnt/β-catenin activity and could be re-sensitized to CRT after pathway inhibition. Other pathways, such as IL-6/JAK/STAT signaling, were similarly shown to mediate CRT resistance in a subset of esophageal adenocarcinomas [60]. Here, multimodal stratification will help to tailor precise therapies for esophageal cancers. Although further studies will be needed to define esophageal cancers with high basal Wnt/β-catenin activity, they may then benefit from Wnt/β-catenin inhibitor-based CRT, our data point to a therapeutic concept with clinical potential.

## 4. Materials and Methods

### 4.1. Cell Lines and Cell Culture

Human esophageal cancer cell lines FLO-1, OAC-P4C, OE-19, OE-33, SK-GT-4 (all from adenocarcinoma), and Kyse-70, Kyse-150, Kyse-180, and Kyse-270 (all from squamous cell carcinoma) were obtained in 2013 directly from the Leibniz Institute DSMZ-German Collection of Microorganisms and Cell Cultures GmbH (DSMZ, Braunschweig, Germany). The DSMZ ensures authenticity of these cell lines using short tandem repeat profiling [61]. Kyse cell lines were established by Shimada et al. [40]. After arrival, all cell lines were expanded and frozen down in aliquots. Cells were cultured in their recommended media (Invitrogen, Carlsbad, Germany), supplemented with 5% or 10% fetal bovine serum (Pan, Aidenbach, Germany), and 2 mM l-glutamine (BioWhittaker, Verviers, Belgium). For experimental use, cells older than 15 passages were discarded. Periodically, mycoplasma contamination was excluded using the MycoAlert^®^ Mycoplasma Detection Kit (Lonza, Cologne, Germany).

### 4.2. Western Blot Analysis

Western blot analysis was performed as previously described [23,24,62]. Briefly, cells were lysed in NP-40 whole cell lysis buffer, and 20 µg of protein was loaded and resolved on a 10% bis-tris polyacrylamide gel. Protein transfer was performed by semi-dry blotting onto a polyvinylidene difluoride membrane (PVDF, GE Healthcare, Little Chalfont, UK), followed by antibody incubation and detection by the ImageQuant LAS 4000 mini CCD camera system (GE Healthcare). Appendix A includes the corresponding antibodies and experimental conditions. Original blot images and calculated band intensities (ImageJ software, version 1.52a, Wayne Rasband, National Institutes of Health, Bethesda, MD, USA) are provided in Appendix A.

### 4.3. Dual Luciferase Reporter Assay

Plasmid transfections were performed as described before [23]. Briefly, for determination of basal Wnt/β-catenin activity, cells were transfected with the reporter plasmids SuperTOP*Flash*, SuperFOP*Flash* (TOP: #12456, FOP: #12457, Addgene, Cambridge, MA, USA), and *Renilla* (Promega, Madison, WI, USA) using X-tremeGENE HP DNA Transfection Reagent (Roche, Penzberg, Germany). To measure the inducibility of the pathway, mutated pCl-neo-β-catenin-S33Y was co-transfected. pCI-neo-β-catenin-S33Y was a gift from Bert Vogelstein (Addgene plasmid # 16519, http://n2t.net/addgene:16519, accessed on 16 August 2021). Twenty-four hours after transfection, cells were lysed by passive lysis buffer (Promega), and both firefly and *Renilla* luciferase activity was measured in a microplate reader (Mithras LB940, Berthold Technologies, Bad Wildbad, Germany). Relative basal transcriptional activation was calculated by dividing *Renilla*-normalized values of SuperTOP*Flash* and SuperFOP*Flash*, whereas inducible activity was calculated by dividing samples that were co-transfected with pCl-neo-β-catenin-S33Y. Detailed experimental conditions are shown in Appendix A.

### 4.4. Cellular Viability Assay

Cellular viability following 5-FU treatment, synthetic small interfering RNA (siRNA)-transfections, or inhibitor treatment (XAV-939, JW55) was assessed using the CellTiter-Blue^®^ reagent (Promega). Reduction of resazurin to resorufin was measured at various time points after the respective treatment using a plate reader (VICTOR™ X4, Perkin Elmer, Waltham, MA, USA) according to the manufacturer’s instructions. Cellular viability of treated or siRNA-transfected cells was compared to untreated cells, or cells transfected with a non-silencing control siRNA (siCtrl.). Detailed information can be found in Appendix A.

### 4.5. siRNA Transfection

Transfections with siRNA duplexes were performed as previously described [62]. Briefly, for cellular viability assays, cells were reverse transfected with siRNA (Qiagen, Hilden, Germany; Dharmacon/Thermo Fisher Scientific, Schwerte, Germany) using RNAiMAX (Invitrogen) or HiPerFect (Qiagen). For colony formation assays and Western blot analyses, cells were transfected using nucleofector technology (Lonza). Additional information about transfection conditions and siRNA sequences can be found in Appendix A.

### 4.6. Chemoradiotherapy and Colony Formation Assays

To test the sensitivity to CRT, standard CFA were conducted as previously described [23,24,62]. Briefly, tumor cells growing in log-phase were seeded as single-cell suspensions into six-well plates. Eight hours after seeding, cells were treated by 3 µM 5-FU (Sigma-Aldrich, Steinheim, Germany), incubated overnight, and subsequently irradiated with single doses of 1, 2, 4, 6, and 8 Gy of X-rays (Gulmay Medical, Camberley, UK). To test the influence of different treatments, cells were either transfected with siRNA, or exposed to inhibitors before irradiation. For irradiation experiments in a fractionated setting, cells were repeatedly irradiated with 2 Gy every 12 h, until a total dose of 10 Gy was reached (Appendix A). After colony formation in the control wells, cells were fixed with 70% ethanol, stained with Mayer’s hemalum solution (Merck KGaA, Darmstadt, Germany), and counted. Colonies were analyzed according to Franken et al. [29]. For a comprehensive evaluation of the effects of the respective treatments (siRNA, inhibitors), a radiation enhancement ratio (RER) was calculated to illustrate the magnitude of radiation sensitization. The RER is defined as the ratio of survival fractions (SF) without and with treatments for a specific dose [39,63]. All experiments were performed in technical triplicates, and independently repeated at least three times (biological replicates). Appendix A shows all experimental conditions for irradiation experiments.

### 4.7. Statistical Analysis

Statistical analyses of SF6 levels for RT and CRT, cellular viability, and luciferase reporter activity experiments were performed using an unpaired two-tailed Student’s *t*-test in Microsoft Excel and visualized in Grapher (version 8.2.460). *p*-values < 0.05 were scored as significant. For analyses of the irradiation data, analysis of variance (ANOVA) was used to calculate significant differences between the control group and treatment group. All analyses were performed using Microsoft Excel software Add-in “Data Analysis” (ANOVA: Two-Factor with Replication). For visualization, irradiation data are presented as mean and standard error of the mean (SEM) from at least three independent experiments using the software KaleidaGraph (version 4.1.0). Again, *p*-values < 0.05 were considered significant, suggesting an influence of the treatment on the dose response. All *p*-values determined in this study are provided in Appendix A.

## Figures and Tables

**Figure 1 ijms-22-10301-f001:**
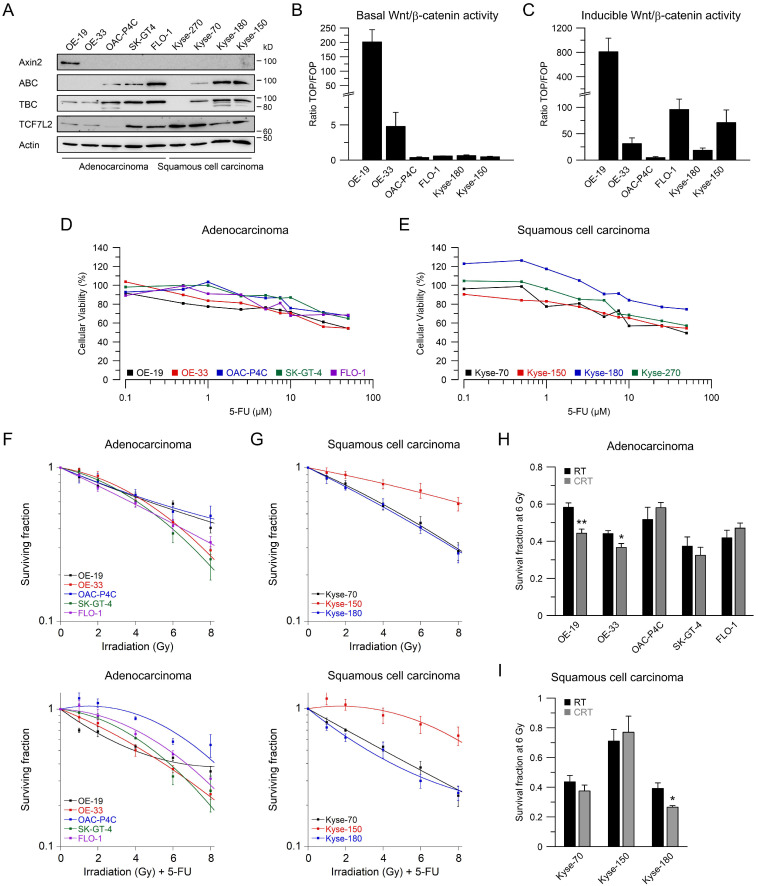
Characterization of Wnt/β-catenin pathway activity, and irradiation sensitivity of esophageal cancer cell lines. (**A**) Nine esophageal cancer cell lines were analyzed for expression of Wnt-related proteins by immunoblotting; (**B**,**C**) Selected cell lines were analyzed for basal Wnt/β-catenin transcriptional activity, (**B**) or inducible transcriptional activity (**C**), measured by dual luciferase reporter assays; (**D**,**E**) Dose-response analysis for different concentrations of 5-FU 24 h after treatment in adenocarcinoma (**D**) or squamous cell carcinoma cell lines (**E**); (**F**,**G**) Adenocarcinoma (**F**) or squamous cell carcinoma cell lines (**G**) were cultured in colony formation assays (CFA) to determine their survival following irradiation (RT, upper panel) or irradiation in the presence of 5-FU (CRT, lower panel); (**H**,**I**) Comparison of RT and CRT survival fractions at 6 Gy. Data are presented as mean ± s.e.m. from at least *n* = 3 independent biological replicates. * *p* < 0.05, ** *p* < 0.01, unpaired two-sample Student’s *t*-test. For *p*-values see Appendix A.

**Figure 2 ijms-22-10301-f002:**
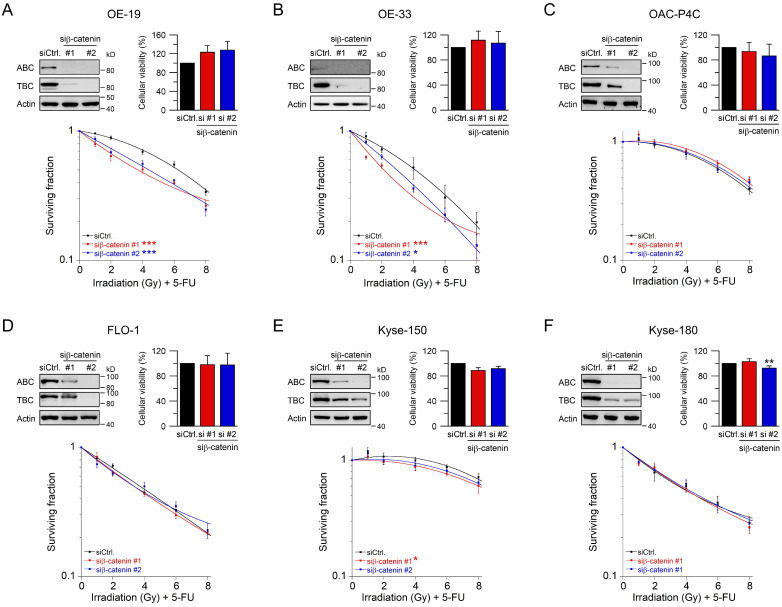
Treatment sensitization following β-catenin depletion depends on the Wnt/β-catenin-pathway activity. (**A**–**F**) Signaling-active cell lines OE-19 (**A**) and OE-33 (**B**), and signaling-inactive cell lines OAC-P4C (**C**), FLO-1 (**D**), Kyse-150 (**E**), and Kyse-180 (**F**) were transfected with control siRNA (siCtrl.) or siRNA targeting *β-catenin* (siβ-catenin #1, #2), and subjected to immunoblot analyses (upper left), and cellular viability assays (upper right). Following siRNA-mediated silencing of β-catenin, cells were monitored for CFA survival after irradiation in the presence of 5-FU (CRT) (lower graph). Data presented as mean ± s.e.m. from at least *n* = 3 independent biological replicates. * *p* < 0.05, ** *p* < 0.01, *** *p* < 0.001, unpaired two-sample Student’s *t*-test or two-way analysis of variance (ANOVA). For *p*-values see Appendix A.

**Figure 3 ijms-22-10301-f003:**
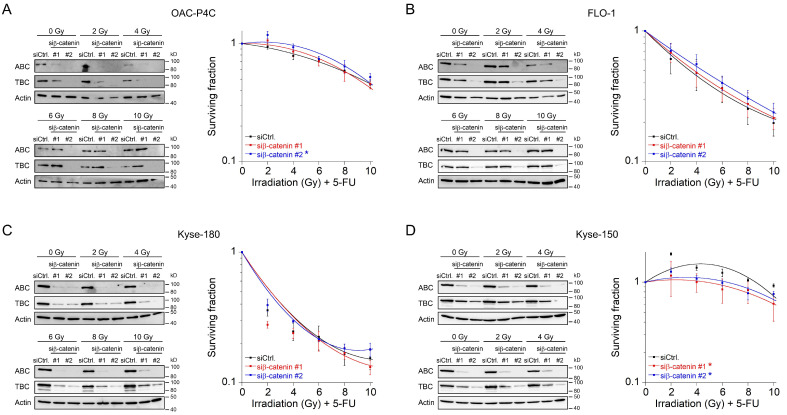
Survival analyses of basal Wnt/β-catenin signaling inactive cells in a fractionated irradiation experimental design. (**A**–**D**) Cell lines OAC-P4C (**A**), FLO-1 (**B**), Kyse-180 (**C**), or Kyse-150 (**D**) were transfected with a control siRNA (siCtrl.), or with siRNA targeting *β-catenin* (siβ-catenin #1, #2), and subjected to immunoblot analyses (left), or monitored for CFA survival after CRT as fractionated irradiation in doses of 2 Gy every twelve hours (right). Data presented as mean ± s.e.m. from at least *n* = 3 independent biological replicates. * *p* < 0.05, two-way analysis of variance (ANOVA). For *p*-values see Appendix A.

**Figure 4 ijms-22-10301-f004:**
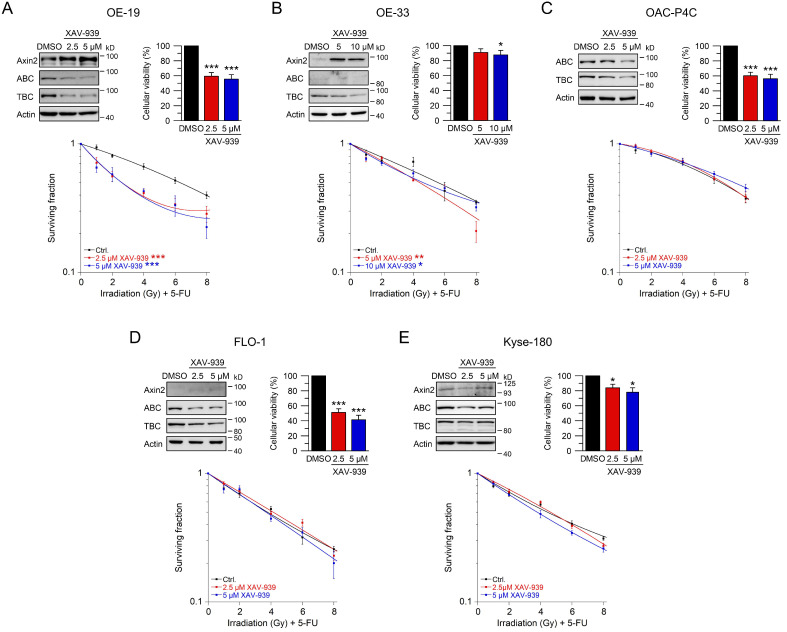
Inhibition of Wnt/β-catenin signaling by the tankyrase inhibitor XAV-939. (**A**–**E**) Cell lines were left untreated (DMSO), or treated with two different concentrations of the tankyrase inhibitor XAV-939. Cell lines ((**A**)—OE-19, (**B**)—OE-33, (**C**)—OAC-P4C, (**D**)—FLO-1, (**E**)—Kyse-180) were first subjected to Western blot analysis (upper left), or cellular viability assays (upper right), or were monitored for CFA survival after CRT (lower graphs). Data presented as mean ± s.e.m. from at least *n* = 3 independent biological replicates. * *p* < 0.05, ** *p* < 0.01, *** *p* < 0.001, two-way analysis of variance (ANOVA). For *p*-values see Appendix A.

**Figure 5 ijms-22-10301-f005:**
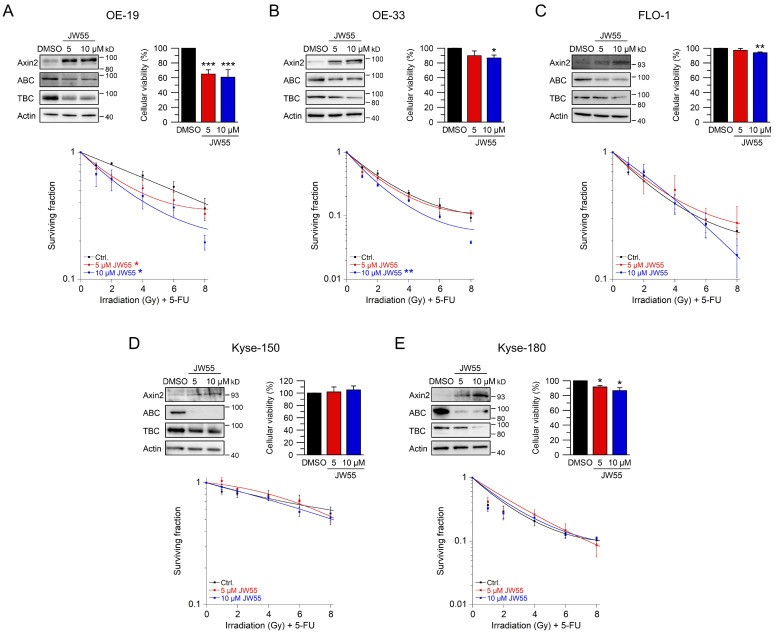
Inhibition of Wnt/β-catenin signaling by the tankyrase inhibitor JW55. (**A**–**E**) Cell lines were left untreated (DMSO), or treated with two different concentrations of the tankyrase inhibitor JW55. Cell lines ((**A**)—OE-19, (**B**)—OE-33, (**C**)—FLO-1, (**D**)—Kyse-150, (**E**)—Kyse-180) were first subjected to Western blot analysis (upper left), or cellular viability assays (upper right), or were monitored for CFA survival after CRT (lower graphs). Data presented as mean ± s.e.m. from at least *n* = 3 independent biological replicates. * *p* < 0.05, ** *p* < 0.01, *** *p* < 0.001, two-way analysis of variance (ANOVA). For *p*-values see Appendix A.

## Data Availability

Data generated can be found in this publication or requested from the corresponding author.

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
