# Peer review of "Inhibition of Wnt/β-Catenin Signaling Sensitizes Esophageal Cancer Cells to Chemoradiotherapy"

_ijms, 2021, doi:10.3390/ijms221910301_

Round 1
Reviewer 1 Report
General comments
The authors in this manuscript provide the experimental data on the Inhibition of Wnt/β-catenin signaling sensitizes esophageal cancer cells to chemoradiotherapy.
Results clearly showed the role of inhibition of this important signaling in sensitizing the esophageal cancer cells to chemoradiotherapy.
The manuscript to me is, in general, clearly written. The science and technical execution of the study is of good quality. The study is solid and the data, in general, support the conclusions. The theory, logic, and experimental design are easy to follow and in general, make sense.
Specific comments
What is the novelty of this study? some recent reports indicate that Wnt signaling induces radioresistance through upregulating HMGB1 in esophageal squamous cell carcinoma (https://www.nature.com/articles/s41419-018-0466-4).
In Fig.1, the authors should add the Wnt/β-catenin activity of the normal cell lines.
Fig. 1.: add the significance.
Fig. 2.: the bottom margin is not clear. Add the significance.
Fig. 3.: Add the significance.
Overall, I believe the improved version of the manuscript will be of interest to the field of oncology. Therefore, it should be recommended for publication in IJMS after revision
Author Response
We thank reviewer 1 for taking her/his time to critically evaluate our manuscript and for the comments, which are highly appreciated.
What is the novelty of this study? Some recent reports indicate that Wnt signaling induces radioresistance through upregulating HMGB1 in esophageal squamous cell carcinoma (https://www.nature.com/articles/s41419-018-0466-4).
We thank the reviewer very much for this comment. We already included this reference into the discussion of our manuscript (reference #48). In the revised version of the manuscript, we extended this part of the discussion to underscore the impact, relevance and novelty of our work.
In Fig.1, the authors should add the Wnt/β-catenin activity of the normal cell lines.
It is a very good advice of this reviewer to add normal cell lines into this analysis. We have currently no normal esophageal epithelial cell line available in our laboratory. However, in a previous publication of our group, we investigated Wnt/β-catenin signaling in colorectal cancer cell lines (Emons et al. Mol Cancer Res. 2017 Nov;15(11):1481-1490). In this paper, we also analyzed a “normal” epithelial cell line, RPE-1. In RPE-1 cells, we observed no basal but inducible Wnt/β-catenin activity (Emons et al. Figure 3A). Because these results are already published, we did not include the activity levels of the normal cell line RPE-1 in this manuscript, but we cite the manuscript accordingly (Reference: Emons et al. Mol Cancer Res. 2017 Nov;15(11):1481-1490).
Fig. 1.: add the significance.
Since we only characterized Wnt/β-catenin activity (Figures 1B and 1C), the influence of 5-FU to cellular viability (Figures 1D and 1E), and the response rates to RT and CRT (Figures 1F and 1G) in esophageal cancer cell lines, no statistical analyses were necessary for these experiments. For analyses shown in Figures 1H and 1I, we have incorporated the significance levels in these panels, as suggested by the reviewer. In the revised version of our manuscript, we exchanged Figure 1, adapted the respective figure legend, and added the respective p-values in Table S1.
Fig. 2.: the bottom margin is not clear. Add the significance.
Fig. 3.: Add the significance.
All figures have been replaced, and now contain the bottom margins now. Asterisks in Figure 2 and Figure 3 represent significance values. In the figure legend, the significance levels are described, and there is also a link to Table S1, which contains the exact p-values of this study.
Reviewer 2 Report
The manuscript is very interesting for the field. The study design seems appropriate and the results well explained. I only recommend that the authors do a last review in the language and typos since some errors were found.
For example:
line 87: please correct “histological different types” to “different histological types”;
Author Response
We thank reviewer 2 for taking her/his time to critically evaluate our manuscript.
We have revised the manuscript for language and removed typos.